# The Application of Multi-Walled Carbon Nanotubes in Bone Tissue Repair Hybrid Scaffolds and the Effect on Cell Growth In Vitro

**DOI:** 10.3390/polym11020230

**Published:** 2019-02-01

**Authors:** Jie Xu, Xueyan Hu, Siyu Jiang, Yiwei Wang, Roxanne Parungao, Shuangshuang Zheng, Yi Nie, Tianqing Liu, Kedong Song

**Affiliations:** 1State Key Laboratory of Fine Chemicals, Dalian R&D Center for Stem Cell and Tissue Engineering, Dalian University of Technology, Dalian 116024, China; xujie172@mail.dlut.edu.cn (J.X.); Huxueyan@mail.dlut.edu.cn (X.H.); syjiang0226@mail.dlut.edu.cn (S.J.); 2Burns Research Group, ANZAC Research Institute, Concord, University of Sydney, Sydney, NSW 2139, Australia; yiweiwang@anzac.edu.au (Y.W.); rpar4161@uni.sydney.edu.au (R.P.); 3Zhengzhou Institute of Emerging Industrial Technology, Zhengzhou 450000, China; sszheng@ipezz.ac.cn; 4Key Laboratory of Green Process and Engineering, Institute of Process Engineering, Chinese Academy of Sciences, Beijing 100190, China

**Keywords:** MWCNTs, CS/Gel/nHAp, bone tissue engineering, MC3T3-E1 cells

## Abstract

In this study, composite scaffolds with different multi-walled carbon nanotubes (MWCNTs) content were prepared by freeze-drying. These scaffolds were characterized by scanning electron microscope (SEM), energy dispersive spectroscopy (EDS), Fourier transform infrared spectroscopy (FTIR), porosity, hydrophilicity, mechanical strength, and degradation. The MWCNTs scaffolds were structurally sound and had porous structures that offered ample space for adherence, proliferation, and differentiation of MC3T3-E1 cells, and also supported the transport of nutrients and metabolic waste. CS/Gel/nHAp/0.3%MWCNTs scaffolds provided the best outcomes in terms of scaffold porosity, hydrophilicity, and degradation rate. However, CS/Gel/nHAp/0.6%MWCNTs scaffolds were found to support the optimal growth, homogenous distribution, and biological activity of MC3T3-E1 cells. The excellent properties of CS/Gel/nHAp/0.6%MWCNTs scaffolds for the adhesion, proliferation, and osteogenesis differentiation of MC3T3-E1 cells in vitro highlights the potential applications of this scaffold in bone tissue regeneration.

## 1. Introduction

Bone tissue, a tough connective tissue consisting of cells, fibers, and a cell matrix, constitutes the skeletal system of the body. The skeletal system provides the framework for movement, attachment points for soft tissues, protection of internal organs, storage of minerals, and a role in the production of blood cells [1,2,3]. Bone tissue is capable of self-repair and regeneration [4]. However, damage to bone tissue from high-impact trauma may result in inadequate self-repair and the need for clinical intervention [5,6]. The deficiencies and complications of current clinical therapies for bone repair and reconstruction have been reported extensively [6,7]. Autogenous bone grafts [8,9], allogeneic bone grafts [10], and bone substitute grafts [11] are currently used in the clinic to repair bone defects. However, these traditional methods are often limited by their small repair range, long repair time, insufficient bone source, secondary injury, and patient rejection [4,5,6,12].

Bone tissue engineering was first proposed by Genevieve M. Crane in 1995 and has since received considerable attention with the development of new research methods and potential applications [13]. Bone tissue engineering involves constructing biological scaffolds to simulate the microenvironment of bone tissue in vitro and combining the scaffold with seed cells and growth factors following transplantation in vivo [14,15]. Three-dimensional (3D) biological scaffolds can support cell attachment, proliferation, and differentiation, and can influence the structure of the tissue-engineered transplant [16]. Chitosan (CS), a high-performance polysaccharide material, was prepared by the deacetylation of natural chitin [17,18]. Gelatin (Gel), a familiar amphoteric polyelectrolyte, is a degraded derivative of structural collagen [19,20]. Cs and Gel are excellent biomaterials that can be used to simulate the function of the extracellular matrix in vivo [21,22]. Furthermore, their biocompatibility, biodegradability, low immunogenicity and cost, and their ability to support cell adhesion, proliferation, and differentiation [22,23,24,25,26] mean they have significant potential in bone tissue engineering [26,27,28,29,30]. Osteoconductivity is a significant property of biomaterials which chemically bonds to bone and supports bone formation [31]. Hydroxyapatite (HAp), a major component of bone mineral, is an osteoconductive biomaterial [31,32,33] that has been generally used to make scaffolds for bone tissue engineering [34,35,36,37,38,39]. Previously, a chitosan/alginate/SiO_2_ (CS/Alg/SiO_2_) composite scaffold prepared by freeze-drying was demonstrated to have a 3D microporous structure, controllable biodegradation, and improved apatite deposition performance that enhanced bone tissue repair [34]. Similarly, porous microspheres prepared with gelatin and hydroxyapatite were shown to be not cytotoxic, but rather promoted cell proliferation and vasodilation for new bone formation [35]. However, the low porosity and mechanical strength which are features of existing bone tissue scaffolds present challenges in bone tissue engineering that need to be addressed.

Carbon nanotubes (CNTs) consist of single-walled carbon nanotubes (SWCNTs) and multi-walled carbon nanotubes (MWCNTs) and are characterized by their high elastic modulus, porosity and mechanical strength, small size, and good electrical and thermal conductivity [40]. CNTs have a surface roughness and surface area similar to collagen fibers in the extracellular matrix, which can greatly promote cell attachment, proliferation, and differentiation [41,42]. In recent years, the MC3T3-E1 cell, an osteoblast cell from a mouse, has been generally used as a cell source for osteogenic differentiation [31,32,38,43]. In this study, CS/Gel/nHAp/MWCNTs scaffolds were prepared by freeze-drying, and the impact of varying MWCNT content on the physicochemical properties and biocompatibility of the scaffold was investigated. Furthermore, MC3T3-E1 cells were inoculated on the CS/Gel/nHAp/MWCNTs scaffolds to assess cell proliferation, differentiation, and mineralization, and to more widely to explore its potential applications in tissue engineering.

## 2. Materials and Methods

### 2.1. Materials

We purchased multi-walled carbon nanotubes (MWCNTs) (diameter 7–11 nm, length 10–30 μm, >90% purity) from Beijing Boyu Gaoke New Materials Technology Co., Ltd. (Beijing, China). MC3T3-E1 cells were provided by the Cell Bank of the Chinese Academy of Sciences. Chitosan (CS, with the degree of deacetylation greater than 90%), gelatin (Gel, analytical reagent, with a purity greater than 99.5%), nanohydroxyapatite (nHAp), and silver nitrate were obtained from Beijing Coolaber Science & Technology Co., Ltd. (Beijing, China). Morpholine ethane sulfonic acid (MES), 1-(3-dimethylaminopropyl)-3-ethylcarbodiimide hydrochloride (EDC), and *N*-hydroxy-succinamide (NHS) were purchased from Shanghai Macklin Biochemical Technology Co., Ltd. (Shanghai, China). Cell Counting Kit-8 (CCK-8), Hoechst 33258, Calcein-AM, and Propidium Iodide (PI) were obtained from Sigma-Aldrich Inc. (St. Louis, MO, USA). The Alizarin Red S reagent kit and Alkaline phosphatase (ALP) assay kit were obtained from Nanjing Jiancheng Bioengineering Institute (Nanjing, China). α-Dulbecco’s Modified Eagle’s Medium (α-DMEM), penicillin-streptomycin, and fetal bovine serum (FBS) were obtained from Hyclone Lab (Logan, UT, USA). All of the above solvents and reagents were of analytical grade.

### 2.2. Preparation of CS/Gel/nHAp/MWCNTs Composite Scaffolds

The preparation of CS/Gel/nHAp and CS/Gel/nHAp/MWCNTs composite scaffolds is shown in Figure 1. A 2.5% solution of CS in 2% aqueous acetic acid was prepared, followed by the addition of Gel particles and nHAp powder into this solution, which was then fully stirred to make a CS/Gel/nHAp mixed solution. This mixed solution was transferred to a 24-pore plate after defoaming in the oven for 30 min and freeze drying for 30 h. Two mL of 65% ethanol aqueous solution containing 50 mmol/L EDC, 50 mmol/L NHS, and 50 mmol/L MES was then added to each hole as a crosslinking agent for 24 h. The acetic acid in the scaffold was neutralized with 0.1 mol/L Na_2_HPO_4_, and the residual crosslinking agent on the scaffold was cleaned with deionized water. Then the CS/Gel/nHAp scaffold was obtained by freeze-drying.

To prepare a CS/Gel/nHAp/MWCNTs scaffold, an MWCNTs solution was first prepared by adding 0.3 g, 0.6 g, or 0.9 g MWCNTs into 100 mL of trace acid solution, and ultrasound dispersed it evenly. MWCNTs acetic acid dispersion solution with a mass volume fraction of 0.3%, 0.6%, and 0.9% were added into the CS/Gel/nHAp mixed solution before cross-linking of the scaffold, and freeze-drying was used to prepare a CS/Gel/nHAp/MWCNTs composite scaffold with an MWCNTs content of 0.3%, 0.6%, and 0.9%.

### 2.3. Characterization of CS/Gel/nHAp/MWCNTs Composite Scaffolds

As 3D tissue scaffolds require appropriate physical and biological properties, the surface morphology, pore structure, chemical composition, porosity, hydrophilicity, mechanical strength, and degradation of CS/Gel/nHAp/MWCNTs composite scaffolds were assessed [44]. The morphological characteristics of the CS/Gel/nHAp/MWCNTs composite scaffolds were observed using a scanning electron microscope (SEM, S2520, Hitachi, Tokyo, Japan). The center of these scaffolds was selected for EDS semi-quantitative analysis using an elemental analyzer (VarioELIII, Elementar, Langenselbold, Germany) to determine the doping of MWCNTs. The specific chemical functional groups of CS/Gel/nHAp and CS/Gel/nHAp/MWCNTs composite scaffolds were tested using a Fourier Transform Infrared Spectroscopy (FTIR, Nicolet 6700, Thermo Fisher, Waltham, MA, USA).

To determine scaffold porosity, a 5 mm × 5 mm × 5 mm scaffold sample was placed into a measuring cylinder containing *V*_1_ of ethanol and placed into a vacuum oven to be subjected to vacuum for 30 min. The total volume of the solution and the scaffold was denoted as *V*_2_. The scaffold immersed in ethanol solution was then taken out of the measuring cylinder, and the volume of remaining ethanol in the measuring cylinder was denoted as *V*_3_. The porosity of the scaffolds was calculated using the following equation:
(1)$$P(\%)=\frac{V_{1}{-V}_{3}}{V_{2}{-V}_{3}}\times 100\%$$


The contact angles of the composite scaffolds with water at a tableting process of 10 Mpa for 1 min were measured using a contact angle meter (OCAH200, Dataphyscics, Filderstadt, Germany). Water absorption was first determined by weighing the composite scaffold in a dry state (*m*_0_). The scaffold was then placed in phosphate buffer solution (PBS) at 37 °C and left to soak for 6 h before the scaffold was bloated against filter paper, and the wet weight (*m*_t_) was measured. The water absorption rate of the scaffolds was calculated using the following equation:
(2)$$A(\%)=\frac{m_{t}{-m}_{0}}{m_{0}}\times 100\%$$


The scaffold was cut into a regular cylinder shape, and the radius and height of the scaffold were measured as *r* and *h*, respectively. The elastic modulus of the scaffold was measured by an electronic universal testing machine (5965, Instron, Norwood, MA, USA) with 1 mm/min of falling speed of pressure head. The elastic modulus of the scaffolds were calculated using the following equation:
(3)$$E=\frac{\varepsilon }{\sigma }=\frac{10\left(L_{2}-L_{1}\right)/S}{\left(D_{2}-D_{1}\right)/h}$$


In the formula, *L*_1_ and *L*_2_ are the pressure loads before and after the linear segment starts, *S* and *h* represent the cross-sectional area and height of the scaffold, and *D*_1_ and *D*_2_ represent the displacement before and after the compression of the scaffold.

To measure the degradation rate of the scaffold, the scaffold was cut into an 8 mm × 8 mm × 8 mm cuboid and was weighed (*m*_0_). The scaffold sample was then placed into a 24-pore plate with 2 mL of lysozyme degradation solution and was left in an incubator at 37 °C for 4 weeks. At the same time every week, three parallel samples were freeze-dried to remove the lysozyme degradation solution and were subsequently weighed (*m*_1_). The degradation rates of the scaffolds were calculated using the following equation:
(4)$$D\left(\%\right)=\frac{\left(m_{0}-m_{1}\right)}{m_{0}}\times 100\%$$


### 2.4. In Vitro Mineralization Ability Test of MC3T3-E1

The 12th generation of MC3T3-E1 cells was cultured in complete medium containing α-DMEM, 10% FBS, and penicillin-streptomycin in a humidified 5% CO_2_ atmosphere at 37 °C for 4 days. The morphology of the MC3T3-E1 cells was observed following no staining, Calcein-AM staining, and Hematoxylin-Eosin (HE) staining using a phase-contrast microscope (IX70, Olympus, Tokyo, Japan). A MC3T3-E1 cell suspension of 2 × 10^6^ cells/mL was cultured with induction medium containing α-DMEM, 10% FBS, penicillin-streptomycin, β-sodium glycerophosphate, and vitamin C. The osteogenic differentiation ability of MC3T3-E1 cells was then ascertained following Alizarin Red S staining (after incubation for 3 weeks), ALP staining (1 week) and silver nitrate (Von-Kossa) staining (4 weeks) by phase-contrast microscopy.

### 2.5. Inoculation and Culture of MC3T3-E1 Cells on Scaffolds

The composite scaffold was cut into thin 5 mm × 5 mm × 1 mm slices and was soaked in 75% ethanol solution and UV irradiated for 12 h. The scaffold was left to soak twice in PBS buffer solution for 2 h, then MC3T3-E1 cells were inoculated onto the scaffold. The MC3T3-E1 cell suspension was prepared using trypsin, and a blood counting chamber was used to regulate cell density to 1 × 10^7^ cells/mL. The cells-scaffold composites were cultured in complete medium containing α-DMEM, 10% FBS, and penicillin-streptomycin in a humidified 5% CO_2_ atmosphere at 37 °C. Complete medium was added regularly, and cell growth was continually observed under microscopy.

### 2.6. Characterization of Cells-Scaffold Composites

After 5 days, 1 mL of 25% glutaraldehyde was added and left overnight on the pore plate of the cells-scaffold composites, and was subsequently fixed with ethanol for 30 min, then the cell morphology was observed by SEM. Then 500 μL of CCK-8 solution containing 1:10 of CCK-8 and α-DMEM was added to the cells-scaffold composites cultured to day 1, 3, 5 and 7 in each pore, and cultured in an incubator for 3 h. After uniform mixing with a pipette, 100 μL of the solution was taken from each pore, and its optical density (OD) at 450 nm was measured using a full wavelength scanning fluorescence spectrometer (Varioskan Flash, Thermo Fisher, USA). The number of living cells was characterized by the OD value.

Cells-scaffold composites cultured to day 2 were washed in PBS solution 3 times and were transferred to a 96-pore plate containing 65 μL tri-stain solution which was prepared by mixing 1 mL of PBS, 2 μL of calcein, 1 μL of PI, and 5 μL of Hoechst33258. Cells-scaffold composites were then incubated in an incubator for 15 min and were washed in PBS solution before being observed under a single photon fluorescence microscope (IX71, Olympus, Japan). In vitro mineralization ability tests of cells-scaffold composites were performed as described in Section 2.4.

### 2.7. Statistical Analysis

All statistical analyses were performed using OriginPro 8.5 software (OriginLab Corporation, Northampton, MA, USA). All experimental data were performed at least three times and reported as the mean ± standard deviation (SD). Significances were analyzed by one-way ANOVA and *t*-testing. Significant differences were marked with *p* < 0.05 (*), *p* < 0.01 (**), *p* < 0.001 (***).

## 3. Results and Discussion

### 3.1. Characterization of CS/Gel/nHAp/MWCNTs Composite Scaffolds

#### 3.1.1. SEM Analysis

In tissue engineering, the pore structure of the scaffold plays a key role in the performance of the scaffold. The pore structure of the composite scaffolds was observed using scanning electron microscopy (Figure 2A). CS/Gel/nHAp scaffolds (Figure 2A top panel) had abundant interconnected porous structures, which are favorable to intercellular contact and mass transfer. In order to see more clearly, the enlarged views were shown in Figure 2A (bottom panel). As shown in Figure 2A in the bottom panel, the walls of the scaffold pores appeared to be rougher and thicker after the addition of MWCNTs, and the diameter of the pores also decreased slightly as MWCNTs content increased (marked with circles). A rough material surface can promote the wetting effect and increase the contact area between the material and cells to facilitate cell adhesion [45]. However, the thickening of the pore wall may lead to a decrease of porosity, thereby reducing the space available for cell growth. Studies have shown that the optimal aperture range of the bone tissue engineering scaffold is 85–325 μm, which provides sufficient space for the adhesion, growth, and differentiation of osteoblasts [46]. The average pore diameter of CS/Gel/nHAp, CS/Gel/nHAp/0.3%MWCNTs, CS/Gel/nHAp/0.6%MWCNTs, and CS/Gel/nHAp/0.9%MWCNTs scaffold were about 176 μm, 178 μm, 169 μm, and 160 μm, respectively (Appendix A). Therefore, SEM analysis of the pore structure of the scaffolds prepared in this experiment demonstrates that they are suitable for bone tissue engineering application.

#### 3.1.2. EDS Analysis

After observation using SEM, central areas of the scaffold were selected to measure the proportion of C element and to determine whether MWCNTs were incorporated into the scaffold. In general, increasing MWCNTs content in the scaffold resulted in a higher C content. The C atomic percent in the CS/Gel/nHAp scaffold, CS/Gel/nHAp/0.3%MWCNTs scaffold, CS/Gel/nHAp/0.6%MWCNTs scaffold, and CS/Gel/nHAp/0.9%MWCNTs scaffold was found to be 92.12%, 92.30%, 93.01%, and 93.61%, respectively, and C weight percent were 79.57%, 80.34%, 81.73%, and 83.01%, respectively (Figure 2B). These results demonstrated that the MWCNTs added to each scaffold were successfully doped into the scaffolds.

#### 3.1.3. FTIR Analysis

FTIR analysis provides information about the main characteristic functional groups of the scaffold to confirm its synthesis route. Figure 3A shows the infrared spectrum of the CS/Gel/nHAp scaffold. The peak at 1152 cm^−1^ corresponds to an asymmetric stretching vibration of the P–O bond, which is a special absorption peak of nano-hydroxyapatite (nHAp). The peaks at 1543 and 1645 cm^−1^ are assigned to the N–H bond of the amide. The amide bond is generated by the chemical reaction of chitosan (CS) and gelatin (Gel) during the cross-linking process. These results indicate that CS, Gel, and nHAp were successfully doped into the composite scaffold. After adding MWCNT, no new characteristic peaks could be observed (Figure 3B) because the carbon skeleton structure of MWCNTs arranged by sp^2^ hybridization rules without special functional groups [40]. However, with increasing MWCNTs content in the scaffold, the light transmittance of each wave crest increases as a whole. This is because the higher the content of MWCNTs added, the relative content of functional groups represented by each wave crest decreases, causing the absorbance of the scaffold at this wave number to decrease and transmittance to increase.

#### 3.1.4. Porosity, Water Absorption, and Contact Angle

Pores in scaffolds exist when particles in the solution are embedded between the polymer chain and the solvent, which prevents the orderly crystallization of the solvent, leaving a small cavity after sublimation [47]. The numbers of pores present in a scaffold represent potential areas of adhesion and growth space for cells, and ultimately the proliferation and spread of cells in the scaffold [44]. Therefore, scaffold porosity is a good indirect measure of scaffold performance. The porosity of the prepared scaffolds was between 78.17% and 93.69%, among which CS/Gel/nHAp/0.3%MWCNTs had the greatest porosity (Figure 4A). The porosity of the scaffolds, however, gradually decreased with increasing MWCNTs content. As MWCNTs content increased, an increase in the thickness of the scaffold wall also contributed to a decrease in porosity. However, compared to the CS/Gel/nHAp scaffold, the effect of the porosity increase due to the decrease of the CS/Gel/nHAp/0.3%MWCNTs scaffold pore size is greater than the porosity decrease due to the increase of pore wall thickness. That resulted in the porosity of the CS/Gel/nHAp/0.3%MWCNTs scaffold being a little higher than the CS/Gel/nHAp scaffold.

The scaffold transplanted into the body will directly contact with blood or tissue fluid, and cell adhesion depends on the hydrophilicity of the scaffold [48]. The water absorption and surface contact angle of the scaffolds were measured to investigate the hydrophilicity of the composite scaffold. CS/Gel/nHAp/0.3%MWCNTs composite scaffolds had the highest water absorption rate at 946.83 ± 2.67% (Figure 4B). As MWCNTs content increased, the water absorption rate of the composite scaffold decreased. The contact angle of CS/Gel/nHAp/0.3%MWCNTs was the smallest at 74.17 ± 2.02% (Figure 4C_1_,C_2_). When MWCNTs content was greater than 0.3%, the contact angle of CS/Gel/nHAp/MWCNTs increased. MWCNTs are hydrophobic materials, and the addition of MWCNTs increases the hydrophobic characteristics of the scaffold, thus reducing the hydrophilicity of scaffold [42]. The CS/Gel/nHAp/0.3%MWCNTs had the smallest contact angle because it had the greatest porosity and highest water absorption. Studies have shown that the surface of these moderately hydrophobic materials is conducive to cell adhesion and growth [49].

#### 3.1.5. Mechanical Test

The mechanical strength of the scaffolds is also an important index that should be considered especially when composite scaffolds are expected to bear heavy stress loads. The linear component of the force-displacement curve (Figure 5A) demonstrates that the addition of MWCNTs influences the mechanical strength of the scaffold. According to the force-displacement curve and the size of the sample, the elastic modulus was determined for each scaffold (Figure 5B). The addition of MWCNTs significantly enhanced the elastic modulus of the scaffold and it was augmented as the MWCNTs content increased. The elastic modulus of the CS/Gel/nHAp/0.9%MWCNTs scaffold at its maximum was 6.58 MPa, which is approximately 3.4 times that of the elastic modulus of the CS/Gel/nHAp scaffold. These results are consistent with previous studies that had shown that MWCNTs have a positive effect on the mechanical strength of the scaffold [50,51]. Therefore, the mechanical strength of the scaffolds had been enhanced by the addition of the MWCNTs, which would be close to the mechanical properties of bone tissue.

#### 3.1.6. Degradation Test

When new tissue forms, scaffolds must be degraded, absorbed, and metabolized. The degradation rate of the tissue engineering scaffold should match the rate of new tissue formation to meet the conditions required for tissue regeneration [52,53]. Therefore, the degradation rate of tissue engineering scaffolds in vivo is an important index used to assess the performance of a scaffold. The in vitro degradation of each scaffold in lysozyme solution is shown in Figure 5C. The degradation rate of each scaffold gradually increased with time. However, the CS/Gel/nHAp scaffold had the highest degradation rate of greater than 80% after 4 weeks. Interestingly, the degradation rate of the scaffolds decreased as MWCNT content increased. As discussed, increasing the MWCNT content increases the hydrophobicity of the scaffold that influences its degradation. Compared to the CS/Gel/nHAp scaffold, there was no significant difference in the degradation rate of the CS/Gel/nHAp/0.3%MWCNTs scaffold. The large interface between the scaffold and water and the high porosity contributed to the high degradation rate of the CS/Gel/nHAp/0.3%MWCNTs scaffold.

### 3.2. The Morphology and Osteogenic Differentiation of MC3T3-E1 Cells

After 4 days culture of the 12th generation of MC3T3-E1, the cells appeared as long fusiform adherent to the wall and had reached 70% confluence in the culture flask (Figure 6A). Connections between the cells forming a helical cell network could be observed, indicating that the MC3T3-E1 have good proliferative potential in vitro. The morphology of the MC3T3-E1 was observed following Calcein-AM staining (Figure 6B) and Hematoxylin-eosin (HE) staining (Figure 6C). MC3T3-E1 long fusiform shaped cells adhered to the wall and covered vortices at the bottom of the pore plate. Following alizarin red staining of the MC3T3-E1 after 3 weeks of induction culture, a large number of red nodules around MC3T3-E1 could be observed, indicating that a large number of calcium nodules were produced during the differentiation of MC3T3-E1 in vitro, laying a good foundation for the formation of new bone (Figure 6D). After 1 week and 4 weeks of osteogenic culture of the MC3T3-E1, cells were stained with alkaline phosphatase (ALP) and Von-Kossa, respectively (Figure 6E,F). A large number of blue-violet particles (Figure 6E) and black particles (Figure 6F) suggested that the MC3T3-E1 cells have great osteogenic differentiation ability. Taken together, the MC3T3-E1 used in this experiment has strong proliferative and osteogenic potential that makes it suitable for use as a seed cell in bone tissue engineering.

### 3.3. Characterization of Cells-Scaffold Composites

#### 3.3.1. SEM Analysis

Scanning electron microscopy (SEM) was used to observe MC3T3-E1 cells cultured on the scaffolds for 5 days. As shown in Figure 7, MC3T3-E1 cells adhered and distributed on the walls of the scaffolds after culture for 5 days, indicating that the scaffolds support the adherence and proliferation of cells. Compared to the other scaffolds, the interconnections (green arrow) which facilitate the transmission of signals between the cells on the CS/Gel/nHAp/0.6%MWCNTs scaffold were a little more intensive. These results indicate that the addition of MWCNTs did not produce any cytotoxic-like effect. Rather, they promoted cell growth and adhesion. In summary, the scaffolds added with MWCNTs have good biocompatibility, and the CS/Gel/nHAp/0.6%MWCNTs scaffold may be the best for supporting the proliferation of MC3T3-E1 cells.

#### 3.3.2. Fluorescent Staining

Single-photon fluorescence microscopy was used to observe the cells-scaffold composites after 2 days of culture (Figure 8). MC3T3-E1 cells adhered and grew well on each scaffold. The cells were interconnected and were distributed on the scaffold surface and pore wall. PI staining demonstrated good cell viability, and only a few dead cells were presented. From the 3D diagram (Appendix A), cells can be seen in the different layers of the scaffold, indicating that MC3T3-E1 cells had infiltrated well into the scaffold where they adhered and proliferated. These results indicated that MC3T3-E1 cells had good cellular activity on all scaffolds, and all composite scaffolds had good biocompatibility.

#### 3.3.3. Cell Activity Assay

The cell activity of MC3T3-E1 cells cultured on each scaffold was determined after 1, 3, 5, and 7 days of culture (Figure 9). Cell activity in the CS/Gel/nHAp scaffold was highest after 1 and 3 days of culture and was significantly lower in the MWCNT content scaffolds. However, by the 5th and 7th day, cell activity for the 0.3%MWCNTs and 0.6%MWCNTs scaffolds were higher when compared to the CS/Gel/nHAp scaffold. Specifically, the 0.6%MWCNTs scaffold group was the highest. Interestingly, cell activity in the 0.9%MWCNTs scaffold was minimal (Appendix A). This result can potentially be explained by the MWCNTs content that may be too high, causing MWCNTs agglomeration on the scaffold that may be toxic to the cells. Therefore, the addition of a small amount of MWCNTs is optimal for cell proliferation, with the CS/Gel/nHAp/0.6%MWCNTs scaffold possessing good biocompatibility properties to support high cell activity.

#### 3.3.4. In Vitro Mineralization Ability Test

In order to evaluate the osteogenic differentiation ability of MC3T3-E1 on the scaffolds, the cells on the scaffolds were stained with ALP (Figure 10). For all cells-scaffold composites, a large amount of blue-purple staining could be observed compared to the scaffold group. This blue-purple staining is indicative of ALP being secreted by MC3T3-E1 during the in vitro culture and differentiation onto the scaffolds, demonstrating good MC3T3-E1 osteogenic differentiation ability. In addition, as the MWCNTs content in the scaffold increased, the blue-purple staining was more intense, especially on the 0.6% MWCNTs scaffolds. The results indicated that the scaffold with certain MWCNTs was conducive to osteogenic differentiation and in vitro mineralization of MC3T3-E1 cells.

## 4. Conclusions

In this study, CS/Gel/nHAp/MWCNTs composite scaffolds with different MWCNTs contents were prepared using a freeze-drying method, and a series of physical and chemical properties of the scaffold were characterized. FTIR analysis demonstrated that CS, Gel, and nHAp were successfully doped into the composite scaffold and maintained good structural integrity. SEM showed that the composite scaffold had an interconnected porous structure, providing ample space for the adhesion, proliferation, and differentiation of osteoblast cells. Furthermore, micro-pore structures were present on the pore wall of the scaffold, improving the mass transfer ability within the scaffold and promoting the transport of nutrients and metabolic waste [44]. The physical properties of the composite scaffolds with different MWCNTs contents were investigated. Although the addition of MWCNTs decreased the degradation rate of the composite scaffold to some extent, it effectively enhanced the mechanical strength of the composite scaffold. Moreover, a 0.3% MWCNTs content provided the best outcomes for porosity, hydrophilicity, and degradation rate.

MC3T3-E1 cells were inoculated on each scaffold, and cells were shown to adhere and extend well on the scaffold. In the CS/Gel/nHAp/0.6%MWCNTs scaffold, the density of MC3T3-E1 cells was greater compared to any other scaffold. These cells maintained good morphology and activity and possessed cell adhesion and proliferative capacity. Through ALP staining of the cells-scaffold composites, the addition of MWCNTs was found to promote the secretion of ALP, and the scaffold with certain MWCNTs was favorable for the osteogenic differentiation of MC3T3-E1 cells in vitro. Taken together, the CS/Gel/nHAp/0.6%MWCNTs composite scaffold prepared in this study had good mechanical strength, porosity, hydrophilicity, degradation, and biocompatibility, which is conducive to the osteogenic differentiation and mineralization of MC3T3-E1 cells in vitro. It is, therefore, a promising scaffold material with potential applications in the field of bone tissue regeneration.

## Figures and Tables

**Figure 1 polymers-11-00230-f001:**
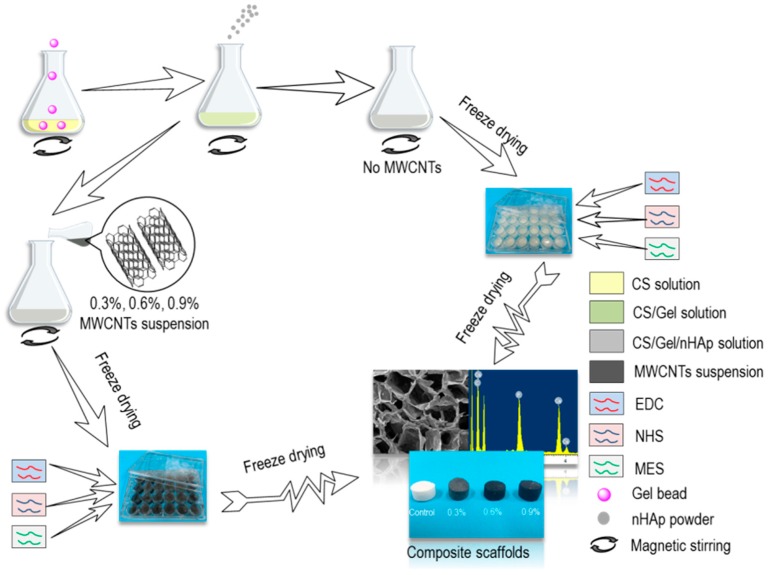
Preparation of CS/Gel/nHAp and CS/Gel/nHAp/MWCNTs composite scaffolds. CS/Gel/nHAp scaffolds were prepared with no MWCNTs, and a CS/Gel/nHAp/MWCNTs scaffold was prepared by adding 0.3%, 0.6%, or 0.9% MWCNTs. Scaffolds were crosslinked with EDC, NHS, and MES as the crosslinking agent. MWCNTs—multi-walled carbon nanotubes; CS—chitosan; Gel—gelatin; nHAp—nano-hydroxyapatite; EDC—1-(3-dimethylaminopropyl)-3-ethylcarbodiimide hydrochloride; NHS—*N*-hydroxy-succinamide; MES—morpholine ethane sulfonic acid.

**Figure 2 polymers-11-00230-f002:**
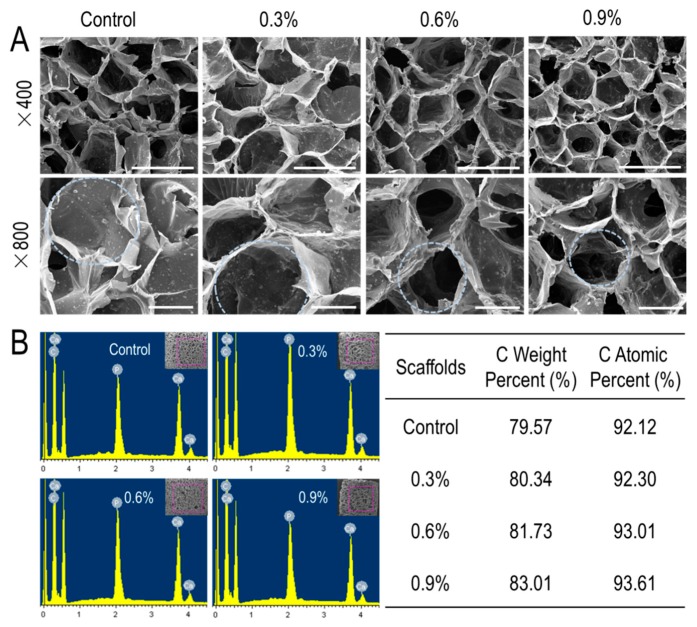
(**A**): SEM observation of CS/Gel/nHAp/MWCNTs scaffolds. ×400 scale: 300 μm; ×800 scale: 100 μm; the circle marks the pores in the scaffold and highlights that the diameter of the scaffold decreased slightly with increasing MWCNTs content. (**B**): Semi-quantitative analysis of CS/Gel/nHAp/MWCNTs scaffolds. Regions of each scaffold were selected for EDS semi-quantitative elemental analysis; the result demonstrates that both the C weight percent and C atomic percent increased with increasing MWCNTs content, and also confirms that MWCNTs were successfully doped into the scaffolds.

**Figure 3 polymers-11-00230-f003:**
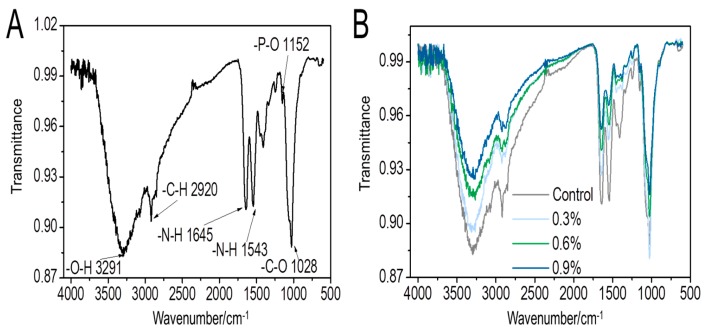
IR spectra of CS/Gel/nHAp/MWCNTs scaffolds. (**A**): Chemical functional group analysis of the CS/Gel/nHAp scaffold; (**B**): IR spectra comparison of each CS/Gel/nHAp/MWCNTs scaffold. The characteristic peaks of the scaffolds did not change; only the transmittance changed with MWCNT content.

**Figure 4 polymers-11-00230-f004:**
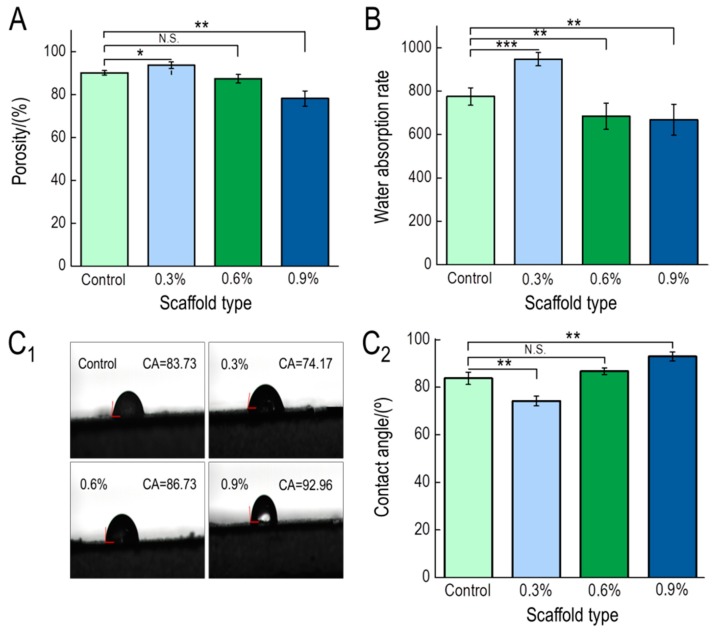
(**A**): Porosity of CS/Gel/nHAp/MWCNTs scaffolds. CS/Gel/nHAp/0.3%MWCNTs scaffold had the greatest porosity. (**B**): Water absorption rate of CS/Gel/nHAp/MWCNTs scaffolds. The CS/Gel/nHAp/0.3%MWCNTs scaffold had the highest water absorption rate. (**C_1_**,**C_2_**): Contact angle of CS/Gel/nHAp/MWCNTs scaffolds. CS/Gel/nHAp/0.3%MWCNTs had the smallest contact angle, indicating that it was the most hydrophilic scaffold. Data indicate mean ± SD. N.S: *p* > 0.05, * *p* < 0.05, ** *p* < 0.01, *** *p* < 0.001.

**Figure 5 polymers-11-00230-f005:**
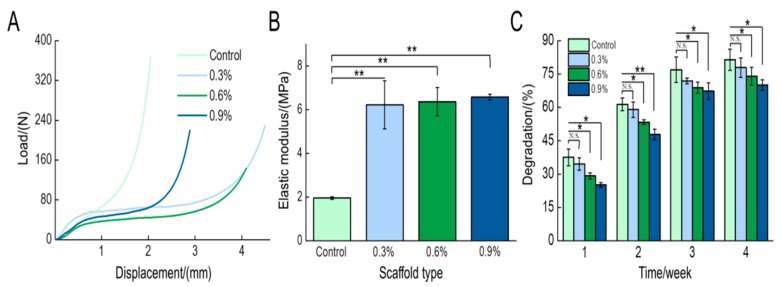
(**A**): The load-displacement curve of the CS/Gel/nHAp/MWCNTs scaffolds. The addition of MWCNTs alters the mechanical strength of the scaffold. (**B**): The elastic modulus of the CS/Gel/nHAp/MWCNTs scaffolds. The addition of MWCNTs significantly increased the elastic modulus of the scaffold. (**C**): The degradation rate of the CS/Gel/nHAp/MWCNTs scaffolds. The degradation rate gradually increased over time but decreased with increasing MWCNTs content at the same time. Data indicate mean ± SD. N.S: *p* > 0.05, * *p* < 0.05, ** *p* < 0.01.

**Figure 6 polymers-11-00230-f006:**
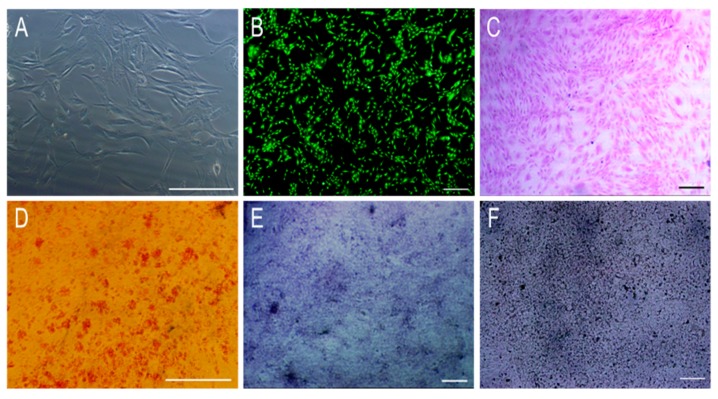
The morphology and differentiation of the 12th generation of MC3T3-E1 cells. (**A**): Cell morphology of the cells; (**B**): Calcein-AM staining of the cells; (**C**): Hematoxylin-eosin (HE) staining of the cells; (**D**): Alizarin red S staining after three weeks of osteogenic induction; (**E**): Alkaline phosphatase (ALP) staining after one week of osteogenic induction; (**F**): Von-Kossa staining after four weeks of osteogenic induction. Scale: A and D: 250 μm. B, C, E and F: 500 μm.

**Figure 7 polymers-11-00230-f007:**
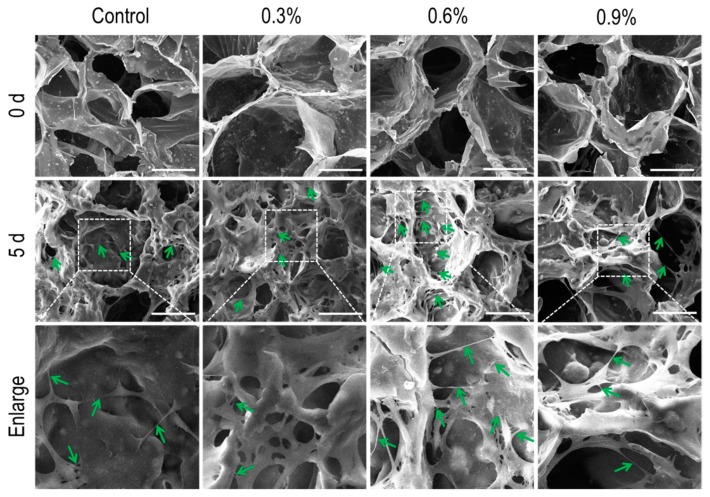
SEM observations of different cells-scaffold composites. The number of cells significantly increased after culture for 5 days compared to 0 days. The white dotted boxes in the middle panel were enlarged in the bottom panel. The green arrow presented the interconnection between the cells, and the interconnection density of the cells cultured on the CS/Gel/nHAp/0.6%MWCNTs scaffold was a little higher compared to other scaffolds. Scale: 100 μm.

**Figure 8 polymers-11-00230-f008:**
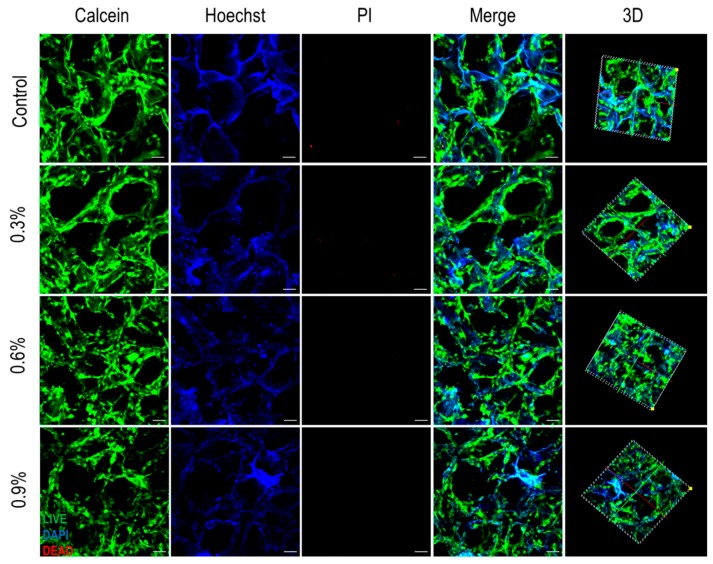
The viability and distribution of MC3T3-E1 cells on CS/Gel/nHAp/MWCNTs scaffolds. Calcein (green) staining showing the spread of live cells, Hoechst (blue) staining showing the cell nucleus, and PI (red) staining showing the spread of the dead cells. The 3D synthesis diagram showing the spread of cells in the 3D structure of the scaffolds. (Incubation for 2 days, scale: 50 μm).

**Figure 9 polymers-11-00230-f009:**
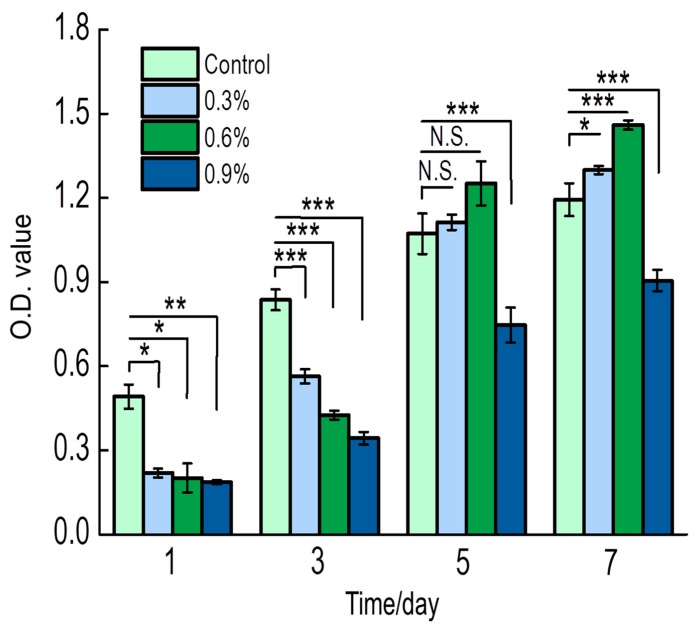
The proliferation of MC3T3-E1 on CS/Gel/nHAp/MWCNTs scaffolds. The cell activity of the CS/Gel/nHAp scaffold was the highest on the 1st and 3rd day, and the cell activity of the MWCNT scaffolds were lower. On the 5th and 7th day, the cell activity of the 0.3% MWCNTs and 0.6% MWCNTs scaffolds were higher than that of the CS/Gel/nHAp scaffold, and the 0.6% MWCNTs scaffold was the highest, but it was lowest for the 0.9% MWCNTs scaffold. Data indicate mean ± SD. The number of living cells was proportional to the OD value. N.S: *p* > 0.05, * *p* < 0.05, ** *p* < 0.01, *** *p* < 0.001.

**Figure 10 polymers-11-00230-f010:**
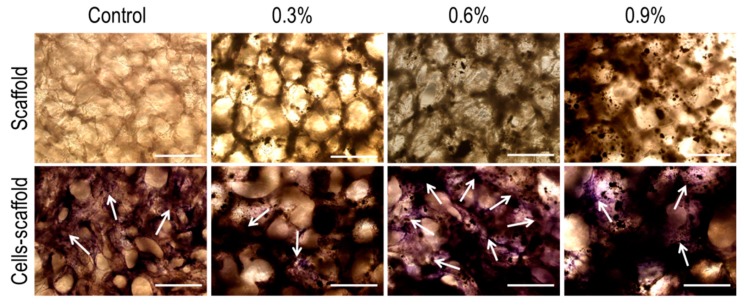
ALP staining of MC3T3-E1 cells on CS/Gel/nHAp/MWCNTs scaffolds. ALP staining (blue-purple) showing the content of ALP that MC3T3-E1 secreted in the process of in vitro culture and differentiation, indicating osteogenic differentiation and mineralization ability of the MC3T3-E1 cells on the scaffolds; the white arrows mark the spread of ALP. Scale: 250 μm.

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
