# Peer review of "The Application of Multi-Walled Carbon Nanotubes in Bone Tissue Repair Hybrid Scaffolds and the Effect on Cell Growth In Vitro"

_polymers, 2019, doi:10.3390/polym11020230_

Round 1
Reviewer 1 Report
The manuscript describes the production, characterisation and in vitro cell growth for hybrid scaffolds fabricated from chitosan, gelatin and hydroxyapatite, and containing a range of concentrations of multi-walled carbon nanotubes. The authors suggest that CS/Gel/nHAp/0.6%MWCNTs are the most favourable scaffold for bone tissue regeneration.
However, I believe that the following points need further attention.
Methods
Assessment of cell metabolic activity (using CCK-8) on scaffolds has been used as a proxy for determining cell number (p 5, line 173). However it is not clear if you are measuring the metabolic activity of cells on the scaffold alone. If some cells do not adhere to the scaffold during seeding they may instead adhere to and continue to grow on the plate underneath. These cells will then still be included in the assessment unless you move the seeded scaffolds to a fresh plate prior to assaying. Please clarify if the scaffold was moved after seeding to ensure that only cells on the scaffold were assayed.
Results and Discussion
2. Figure 2 (p 6). In the text you state “CS/Gel/nHAp scaffolds (Figure 2A top panel) had an abundant interconnected porous structure” (p 5, line 191), and then indicate “and the diameter of the pores also decreased slightly as MWCNTs content increased (Figure 2A bottom panel” (p 5, line 193). This is confusing as the way it is written suggests that the top panel of Figure 2 A is without MWCNTs and the bottom panel contains them, which is clearly not the case. This needs to be written more clearly
3. The manuscript states “The diameter of CS/Gel/nHAP scaffold and CS/Gel/nHAP/MWCNTs scaffolds were found to be between 120 μm and 330 μm” (p 5, line 200). I assume that this is actually referring to pore size diameter rather than scaffold diameter? If so it needs to be clarified. In addition, I would expect to see the data for the mean pore size and distribution for each scaffold type as well as quoting this range for all the samples together, particularly as the manuscript describes a change in pore size for different samples (p 7, line 251).
4. There is a mistake in the data for figures 4 and 5. The legends and data for figures 4 and 5 appear to have been swapped round, and when the text refers to figure 4 (e.g. p 7, line 247) the relevant data is actually to be found in figure 5, and vice versa (e.g. p 8, line 276).
5. The manuscript indicates that “carbon nanotubes disperse uniformly in the scaffold” (p 8, line 284), however it is not clear where the evidence is for this statement
6. Figure 6 shows an assessment of MC3T3-E1 cells cultured on tissue culture plastic. This cell line has been widely used as a model cell line for bone biology (over 150 publications in last 5 years) and there is no requirement to demonstrate that the cell line is suitable for bone tissue engineering. Section 3.2 and figure 6 are therefore not required and should be removed.
7. Figure 7. The legend states that “the number of cells significantly increased from 5 days of culture, compared to 3 days” however from these images it is not possible to determine any level of significant difference, since the data is purely qualitative. It is also difficult to see from these images if cell density is actually highest on the 0.6% MWCNT scaffolds and this statement should be removed.
8. Figure 8. It appears from the images in this figure that scaffold used may be autofluorescent in both green and blue channels. This figure therefore requires an additional control showing the scaffolds without cells imaged using the same imaging settings for comparison. The Hoechst DNA staining in particular does not appear to localise as expected, with significant blue staining in areas that appear to be outside cells. Also it is unclear what depth within the sample these images were obtained from. Finally, the 3D images, as they are shown, give no additional information compared to the merged image, in fact in some cases they look identical to the merged image. If 3D images are to be presented they should be orientated in a manner to allow the reader to view cell distribution throughout the scaffold thickness.
9. Figure 9. The data shown only compares cell growth on the various scaffolds, with and without the addition of MWCNTs. Although the manuscript refers to the scaffold with 0% MWCNTs as the control for cell growth I would have also expected to see a control with the same cells on tissue culture plastic. According to the references you have cited, there has been previous work using a gelatin/chitosan/hydroxyapatite (ref 26) scaffold, however this appears to have been generated using a different method. Therefore cell growth on the hybrid scaffold without MWCNTs appears to be untested, making it unsuitable as a positive control.
10. The manuscript suggests that CS/Gel/nHAp/0.6%MWCNTs scaffold had the most intense ALP staining, with weak staining on CS/Gel/nHAp/0.9%MWCNTs scaffold (p 13, line 385), however an examination of Figure 10 suggests that this claim is hard to substantiate, with staining looking quite similar in all images. Was any attempt made to measure the intensity of staining in a quantitative manner?
Conclusions
11. You state “In the CS/Gel/nHAp/0.6%MWCNTs scaffold, the density of MC3T3-E1 cells that were evenly distributed throughout the scaffold was greater compared to any other scaffold.” (p 14, line 409). The data as shown in this paper does not actually provide any information about distribution in any scaffold. The only conclusion that your data appears to allow you to claim is that cell metabolic activity was greatest on the CS/Gel/nHAp/0.6%MWCNTs scaffold. However, as in comment 1, if this was assessed with scaffolds in the same well in which they were initially seeded the increased growth may be due in part to cells on the tissue culture plastic.
12. The conclusion suggests that micro-pore structures were present on the pore wall of the scaffold, improving the mass transfer ability within the scaffold, and that facilitating the transport of nutrients and metabolic waste (p 14, line 401) however there is no data in the manuscript to substantiate the claim of improved mass transfer, and it can be nothing more than a hypothesis that this may be the case. Also, the suggested micro-pore structures are not identified or discussed earlier in the manuscript and it is unclear what micro-pores the authors are referring to.

Author Response
Response to Reviewer 1 Comments
Point 1: Assessment of cell metabolic activity (using CCK-8) on scaffolds has been used as a proxy for determining cell number (p 5, line 173). However it is not clear if you are measuring the metabolic activity of cells on the scaffold alone. If some cells do not adhere to the scaffold during seeding they may instead adhere to and continue to grow on the plate underneath. These cells will then still be included in the assessment unless you move the seeded scaffolds to a fresh plate prior to assaying. Please clarify if the scaffold was moved after seeding to ensure that only cells on the scaffold were assayed.
Response 1: Many thanks for the suggestion of reviewer. 8 μL of MC3T3-E1 cell suspension was inoculated onto the two sides of the scaffold, and then a little (200 μL) complete medium was slowly added into each pore to nourish the inoculated cells. 500 μL of complete medium each time was slowly added regularly to provide growing environment after cells adhered to the scaffolds. In this process, some cells adhered to and grew on the plate, but this number was tiny and almost all cells were adhered to the hybrid scaffolds. With the above treatment, cell loss has been largely avoided. In order to unify variables, we took 100 μL/500 μL of the solution in each pore (section 2.6) to measure the cell activity. In this way, the data were largely close to the real value that cell activity (cell number) of cells on the scaffolds.
Point 2: Figure 2 (p 6). In the text you state “CS/Gel/nHAp scaffolds (Figure 2A top panel) had an abundant interconnected porous structure” (p 5, line 191), and then indicate “and the diameter of the pores also decreased slightly as MWCNTs content increased (Figure 2A bottom panel” (p 5, line 193). This is confusing as the way it is written suggests that the top panel of Figure 2 A is without MWCNTs and the bottom panel contains them, which is clearly not the case. This needs to be written more clearly
Response 2: Many thanks for the suggestion of reviewer. It not the meaning that the top panel of Figure 2A is without MWCNTs and the bottom panel contains them, but the bottom panel were the enlarged images of the top panel, which could see more clearly the pore diameter of the scaffolds. Following the suggestion of the reviewer, the statement was revised to “In order to see more clearly, the enlarged view were shown in Figure 2A bottom panel. As shown in Figure 2A bottom panel, the wall of the scaffold pores appeared to be rougher and thicker after addition of MWCNTs, and the diameter of the pores also decreased slightly as MWCNTs content increased (marked with circles).” In our revised manuscript R1, we have explained that the wall of the scaffold pores appeared to be rougher which could facilitate cell adhesion as MWCNTs content increased. But on the same time the wall would be thicker which lead to a decrease of the porosity and the diameter of the pores, which reducing the space available for cell growth. Therefore, we should choose the optimal MWCNTs content according to the cell adhesion, proliferation and osteogenesis differentiation of the cells on the scaffolds.
Point 3: The manuscript states “The diameter of CS/Gel/nHAP scaffold and CS/Gel/nHAP/MWCNTs scaffolds were found to be between 120 μm and 330 μm” (p 5, line 200). I assume that this is actually referring to pore size diameter rather than scaffold diameter? If so it needs to be clarified. In addition, I would expect to see the data for the mean pore size and distribution for each scaffold type as well as quoting this range for all the samples together, particularly as the manuscript describes a change in pore size for different samples (p 7, line 251).
Response 3: Many thanks for the great suggestion of reviewer. The diameter is referring to pore size diameter. Following the suggestion of reviewer, we measured the pore diameter of CS/Gel/nHAP scaffold and CS/Gel/nHAP/MWCNTs scaffolds by Image-pro plus software (Supplementary Table 1). The average pore diameters of CS/Gel/nHAP, CS/Gel/nHAP/0.3%MWCNTs, CS/Gel/nHAP/0.6%MWCNTs and CS/Gel/nHAP/0.9%MWCNTs scaffolds were around 176 μm, 178 μm, 169 μm and 160 μm, respectively. To some extent, this is similar to the change tendency of porosity (Figure 4A).
Point 4: There is a mistake in the data for figures 4 and 5. The legends and data for figures 4 and 5 appear to have been swapped round, and when the text refers to figure 4 (e.g. p 7, line 247) the relevant data is actually to be found in figure 5, and vice versa (e.g. p 8, line 276).
Response 4: Many thanks for considerate advice of reviewer. Figure 4 and 5 have been swapped back in the revised manuscript R1.
Point 5: The manuscript indicates that “carbon nanotubes disperse uniformly in the scaffold” (p 8, line 284), however it is not clear where the evidence is for this statement
Response 5: Many thanks for the great suggestion of reviewer. What the statement’s meaning is that “carbon nanotubes have good mechanical properties that contribute to enhancing the mechanical strength of the scaffold”. In order to describe more clearly, the sentence was revised to “Therefore, the mechanical strength of scaffolds had been enhanced by addition of the MWCNTs, which would be close to the mechanical properties of bone tissue”.
Point 6: Figure 6 shows an assessment of MC3T3-E1 cells cultured on tissue culture plastic. This cell line has been widely used as a model cell line for bone biology (over 150 publications in last 5 years) and there is no requirement to demonstrate that the cell line is suitable for bone tissue engineering. Section 3.2 and figure 6 are therefore not required and should be removed.
Response 6: The MC3T3-E1 cells has been widely used as a tipical cell source for osteogenic differentiation in recent study (last paragraph of introduction), so it is very useful in bone tissue engineering, and we used it as a seed cell in our study. But cell generation of the MC3T3-E1 (12th generation in the study) is numerous, so the morphology and osteogenic differentiation ability of the different generation MC3T3-E1 should be re-identified. Moreover, images of Figure 6 also were the pure cell control group of the cells-scaffold composites group, whose morphology and osteogenic differentiation had certain similarity and difference compared to Figure 6. For example, we could discern the MC3T3-E1 cells on the scaffolds according to the morphology of the MC3T3-E1 cells in Figure 6, and we also could detect the difference of degree between the ALP staining of pure cell group and cells-scaffold composites group. Only in this way can the integrity of the research be guaranteed. Considering this point, we think we should keep the section 3.2 and Figure 6.
Point 7: Figure 7. The legend states that “the number of cells significantly increased from 5 days of culture, compared to 3 days” however from these images it is not possible to determine any level of significant difference, since the data is purely qualitative. It is also difficult to see from these images if cell density is actually highest on the 0.6% MWCNT scaffolds and this statement should be removed.
Response 7: Many thanks for the great suggestion of reviewer. The data concered with SEM images are indeed qualitative. Althought the number of cells did not significantly increase after 5 days of culture while compared to the culture point at day 3, but it was confirmed that the cell nubmer significantly increased in 3 and 5 days of culture when compared with day 0. And we could see that the interconnection between the cells on 0.6%MWCNTs scaffold were a little more intensive compared to other kinds of scaffolds. Following the suggestion of reviewer, we removed the images of day 3 and the statement that “the number of cells significantly increased from 5 days of culture, compared to 3 days”. The statement was revised to “Compared to the other scaffolds, the interconnection (green arrow) which facilitate the transmission of signals between the cells on the CS/Gel/nHAp/0.6%MWCNTs scaffold were a little more intensive”.
Point 8: Figure 8. It appears from the images in this figure that scaffold used may be autofluorescent in both green and blue channels. This figure therefore requires an additional control showing the scaffolds without cells imaged using the same imaging settings for comparison. The Hoechst DNA staining in particular does not appear to localise as expected, with significant blue staining in areas that appear to be outside cells. Also it is unclear what depth within the sample these images were obtained from. Finally, the 3D images, as they are shown, give no additional information compared to the merged image, in fact in some cases they look identical to the merged image. If 3D images are to be presented they should be orientated in a manner to allow the reader to view cell distribution throughout the scaffold thickness.
Response 8: Many thanks for the suggestion of reviewer. Calcein staining was used to detect living cells, Hoechst DNA staining was used to mark both living and dead cells, and PI staining was only to detect dead cells. Therefore an additional control staining of the scaffold without cells would be nothing but a completely black image. Calcein staining showed that there were amounts of living cells on the scaffolds, and PI staining shows a few dead cells, so there was no much difference between Calcein and Hoechst DNA staining images but the colour presented.
Based on this, we added the following file (Supplementary Fiugure 1) to the revised manucript R1. From the 3D images (Supplementary Fiugure 1), we could see that MC3T3-E1 cells had well-distributed on the different layers of the hybrid scaffolds. In summary, the aim of Figure 8 was to demonstrate that MC3T3-E1 cells adhered and grew well on these scaffolds and this scaffolds had good biocompatibility.
Point 9: Figure 9. The data shown only compares cell growth on the various scaffolds, with and without the addition of MWCNTs. Although the manuscript refers to the scaffold with 0% MWCNTs as the control for cell growth I would have also expected to see a control with the same cells on tissue culture plastic. According to the references you have cited, there has been previous work using a gelatin/chitosan/hydroxyapatite (ref 26) scaffold, however this appears to have been generated using a different method. Therefore cell growth on the hybrid scaffold without MWCNTs appears to be untested, making it unsuitable as a positive control.
Response 9: Many thanks for the suggestion of reviewer. The Figure 9 quantitatively compared to Figure 7, and it was quantificationally to demonstrate that MC3T3-E1 cells adhered and grew well on the hybrid scaffolds. And the cell activity for the 0.3%MWCNTs and 0.6%MWCNTs scaffolds were higher compared to the CS/Gel/nHAp scaffold (cell growth on the hybrid scaffold without MWCNTs already be tested, it was negative control, but not positive control) after cultured 5 or longer days. As the negative control of the scaffold without MWCNTs and the positive control of the scaffold with 0.3% and 0.9%MWCNTs, the cell actively of 0.6%MWCNTs scaffold group was the highest. In other words, the scaffolds with certain MWCNTs in the manuscript were novel compared to Gel/CS/HAp scaffold (ref 26), so the method was different. In this manuscript, the results show that the mechanical strength of the scaffold was enhanced by MWCNTs, other physcial properties were satisfactory, and moreover cells on CS/Gel/nHAP/0.6%MWCNTs scaffold showed the best cell activity. In order to shown the results more visual, Supplementary Figure 2 data was given as following.
Point 10: The manuscript suggests that CS/Gel/nHAp/0.6%MWCNTs scaffold had the most intense ALP staining, with weak staining on CS/Gel/nHAp/0.9%MWCNTs scaffold (p 13, line 385), however an examination of Figure 10 suggests that this claim is hard to substantiate, with staining looking quite similar in all images. Was any attempt made to measure the intensity of staining in a quantitative manner?
Response 10: Many thanks for the great suggestion of reviewer. Althought the osteogenic differentiation ability of MC3T3-E1 cells on the scaffolds could not be quantitatively decribed, the blue-purple stainings on 0.6%MWCNTs scaffold had significantly increased compared to control and 0.3% groups, and a little weak staining on 0.9%MWCNTs scaffold could be found when compared to 0.6% group. In order to decribe more accurately, the statement that “In contrast, weak ALP staining of cells cultured on the 0.9%MWCNTs scaffold……” had been revised to “In contrast, a little weak ALP staining of cells cultured on the 0.9%MWCNTs scaffold……”.
Point 11: You state “In the CS/Gel/nHAp/0.6%MWCNTs scaffold, the density of MC3T3-E1 cells that were evenly distributed throughout the scaffold was greater compared to any other scaffold.” (p 14, line 409). The data as shown in this paper does not actually provide any information about distribution in any scaffold. The only conclusion that your data appears to allow you to claim is that cell metabolic activity was greatest on the CS/Gel/nHAp/0.6%MWCNTs scaffold. However, as in comment 1, if this was assessed with scaffolds in the same well in which they were initially seeded the increased growth may be due in part to cells on the tissue culture plastic.
Response 11: Many thanks for the suggestion of reviewer. The data as shown in Figure 7, 8, and supplementary Figures 1 & 2 had qualitatively described that MC3T3-E1 cells distributed throughout the CS/Gel/nHAp/0.6%MWCNTs scaffolds, and Figure 9 in revised manuscript R1 had quantificationally described that the number of MC3T3-E1 cells on CS/Gel/nHAp/0.6%MWCNTs scaffolds was the lagrest one while compared to any other scaffolds. That due to the number of living cells was proportional to the OD value.
Point 12: The conclusion suggests that micro-pore structures were present on the pore wall of the scaffold, improving the mass transfer ability within the scaffold, and that facilitating the transport of nutrients and metabolic waste (p 14, line 401) however there is no data in the manuscript to substantiate the claim of improved mass transfer, and it can be nothing more than a hypothesis that this may be the case. Also, the suggested micro-pore structures are not identified or discussed earlier in the manuscript and it is unclear what micro-pores the authors are referring to.
Response 12: Many thanks for the suggestion of reviewer. The micro-pores in the manuscript are referring to micro-pore structures of the hybrid scaffolds, whose diameter were described in Supplementary Table 1. The claim of “micro-pore improving the mass transfer” had been demonstrated and confirmed in other work (ref 44). Following the suggestion of reviewer, we added the reference [44] in the sentence (in Page 14, section 4, 3th sentence, line 407 ) that “Furthermore, micro-pore structures were present on the pore wall of the scaffold, improving the mass transfer ability within the scaffold, and that facilitating the transport of nutrients and metabolic waste”.

Reviewer 2 Report
The current manuscript provides a detailed account of composite scaffolds impregnated with different multi-walled carbon nanotube (MWCNTs) and their application with respect to bone tissue engineering is discussed. I recommend major revisions for the manuscript based on the following comments:
The synthesis and source of MWCNTs is not decribed/mentioned. The type, size, shape, SEM/TEM and other properties of MWCNTs used in the study should be provided.
SEM analysis of Cells-Scaffold Composites: The SEM should essentially be zoomed in to show the cell-surface interactions and morphology.
In vitro mineralization ability test: This should be backed by SEM/EDS analysis. This point is very important and must be addressed.
Author Response
Response to Reviewer 2 Comments
Point 1: The synthesis and source of MWCNTs is not decribed/mentioned. The type, size, shape, SEM/TEM and other properties of MWCNTs used in the study should be provided.
Response 1: Many thanks for the suggestion of reviewer. Carbon nanotubes (CNTs), are mainly coaxial tubes with several or dozens of layers of Carbon atoms arranged in a hexagonal pattern. MWCNTs (diameter 7-11 nm, length 10-30 μm, >90% purity), one kind of CNTs, were the commerically available material and were purchased from Beijing Boyu Gaoke New Materials Technology Co., Ltd. (Beijing, China). Following the suggestion of reviewer, we added the size, shape, and purity information of the MWCNTs that “diameter 7-11 nm, length 10-30 μm, >90% purity” in the section 2.1, line 77. MWCNTs were characterized by their high elastic modulus and porosity, small size, and good electrical and thermal conductivity (ref 40). In our current study, we just used the property of high elastic modulus of the MWCNTs to enhance the mechanical strength of the hybrid scaffolds.
Point 2: SEM analysis of Cells-Scaffold Composites: The SEM should essentially be zoomed in to show the cell-surface interactions and morphology.
Response 2: Many thanks for the great suggestion of reviewer. Following the suggestion of reviewer, we had added the enlarged images of cells-scaffold composites cultured for 5 days to shown the cell-surface interactions and morphology (Figure 7).
Point 3: In vitro mineralization ability test: This should be backed by SEM/EDS analysis. This point is very important and must be addressed.
Response 3: Many thanks for the suggestion of reviewer. In the current revised manuscript, the aim of in vitro mineralization ability test was not to observe the surface morphology of the cells-scaffold composites, but to investigate the osteogenic differentiation ability of the MC3T3-E1 cells within/on the hybrid scaffolds. Moreover, we had already tested the elments of the hybrid scaffolds by EDS analysis in section 3.1.2 and we had observed the cells-scaffold composites by SEM analysis in section 3.3.1, which indicated that we had successfully prepared the hybrid scaffolds and the scaffolds also supported the adherence and proliferation of the MC3T3-E1 cells.

Round 2
Reviewer 1 Report
Comments included as attached file

Author Response
Response to Reviewer 1 Comments
Point 8: I still have concerns over figure 7. Following staining there appears to be a significant levels of blue signal which is outside areas where there are cells i.e. where there appears to be no green (live cells) or red (dead cell) signal. Since the Hoescht stain (blue) is staining the nuclei of cells it should not appear in regions where there are no cells present.
See this section of the figure, where the three signals are merged, as an example: Examples of regions of sample which is stained blue but seems to be outside of an area with cells (i.e. not green or red)
In my initial review I suggested that this may be because the material is itself autofluorescent in the blue channel. In the Hoescht images the blue signal appears to come from two morphologically different areas, these are highlighted as an example in part of the figure below: Region where blue stain appears to be morphologically similar to the scaffold – and therefore may be due to scaffold autofluorescence material; Region where blue stain has the expected, rounded nuclear morphology.
The authors argue that an additional control with no cells would be black, however the evidence within this publication does not substantiate this. The material is clearly black when viewed in the red channel, but there is nothing to show that it would appear black if viewed in either the blue or green channels. In addition, the supplementary figure 2 also seems to confirm my suspicion that the material is autofluorescent in the blue channel, with a significant blue signal appearing even on day 1 when there were very few cells detected by Calcein staining.
As a result I asked that a control image be included that showed the scaffold without cells, obtained using the same imaging settings, for all three channels. This would clearly prove that the images shown were indeed cell nuclei (blue Hoescht stain) or viable cell cytoplasm (green Calcein stain), rather than a combination of autofluorescent scaffold and cells. In my opinion, this control is still required to demonstrate the real level of cell growth on these scaffolds.
Response: Many thanks for the suggestion of reviewer. This is “Point 8 & 9” in the initial review. Following the suggestions of the reviewer, we did a validation experiment to judge if the scaffolds without cells would be autofluorescent in the green, blue and red channel, the result was shown as following images.
As shown in above images, the scaffolds (negative control and 0.6% group) both have intense autofluorescent in blue channel, weaker autofluorescent in green channel and almost invisible autofluorescent in red channel. This results confirmed reviewer’s guess that the scaffolds are kinds of autofluorescent materials. Although these materials presented autofluorescence to a certain degree, we can also clearly observe the specific distribution of the inoculated cells in different scaffolds if the shapes of labeled cell adhesion and extension could be well combined.
As shown in Figure 8, many long fusiform MC3T3-E1 cells (green) adhere onto the scaffolds in Calcein images, and many punctate nucleus (blue) could be observed in Hoechst images. In the Merge images, the distribution of the long fusiform MC3T3-E1 cells (green) was emerged more striking after blue channel autofluorescent covered the green channel. In addition, long fusiform MC3T3-E1 cells (green) were observed from both sides of the scaffolds (due to the thin slice of 5 mm×5 mm×1 mm) in the 3D images (Supplementary Figure 1), which showing they distributed on the different layers of the scaffolds.
Figure 8. The viability and distribution of MC3T3-E1 cells on CS/Gel/nHAp/MWCNTs scaffolds. Calcein (green) and Hoechst staining showing the spread of live cells, and PI (red) staining showing the spread of the dead cells. The 3D synthesis diagram showing the spread of cells in the 3D structure of the scaffolds. (Incubation for 2 days, Scale: 50 μm).
Supplementary Figure 1. The 3D images distribution of MC3T3-E1 cells on different scaffolds. The 3D images show the 3D distribution observed from both sides of the scaffolds (due to the thin slice of 5 mm×5 mm×1 mm), and the MC3T3-E1 cells distributed on the different layers of the scaffolds.
Point 9: I do not understand what is now meant by a “little weak staining of cells cultured on the 0.9%MWCNTs scaffold”. I believe that in the figure below, there little difference in ALP staining intensity between samples. The control and 0.3% samples appear very slightly lighter than the other two, but I do not believe there is any difference in staining intensity seen for the 0.6% and 0.9% samples. If the authors are unable to repeat and quantify staining using an appropriate method, I believe that no attempt should be made to claim that 0.6% staining is more intense than 0.9%, and no conclusion of an improved mineralisation for 0.6% compared to 0.9% can be drawn. Without further evidence, these comments should be removed.
Response: Many thanks for the suggestion of reviewer. This is “Point 10” in the initial review. Following the suggestion of the reviewer, we removed the comment that “In contrast, a little weak ALP staining of cells cultured on the 0.9%MWCNTs scaffold highlighted that this scaffold lacked the conditions to support cell growth and survival. As discussed, excessive MWCNTs content could potentially lead to MWCNT agglomeration and cell toxicity. CS/Gel/nHAp/0.6%MWCNTs scaffold was found to be the most favorable for in vitro mineralization of MC3T3-E1.” And the sentence was revised to “The results indicated that the scaffold with certain MWCNTs was conducive to osteogenic differentiation and in vitro mineralization of MC3T3-E1.”
Point 11: The authors have not removed their comment about cells being evenly distributed, even though they are unable to provide any information which substantiates details regarding the distribution of cells throughout the scaffold. The fluorescent images shown do not make it possible to make any determination of cell distribution within the scaffold and the 3D images still show the view from above so it is not possible to see how much the cells have actually penetrated into the scaffold and whether distribution is even or not. This comment needs to be removed or additional evidence provided.
Response: Many thanks for the suggestion of reviewer. Following the suggestion of the reviewer, we removed information about cells being evenly distributed. In details, we revised “throughtout” in line 336 (section 3.3.1) to “on”, and we removed “evenly” in line 353 (section 3.3.2) and “that were evenly distributed throughout the scaffold” in line 417 (section 4).

Reviewer 2 Report
The authors have addressed the comments raised by the reviewer and the revised manuscript is much improved from the original manuscript.
Author Response
Many thanks to the suggestion of the reviewer 2.